# Serum CD133-Associated Proteins Identified by Machine Learning Are Connected to Neural Development, Cancer Pathways, and 12-Month Survival in Glioblastoma

**DOI:** 10.3390/cancers16152740

**Published:** 2024-08-01

**Authors:** Thomas Joyce, Erdal Tasci, Sarisha Jagasia, Jason Shephard, Shreya Chappidi, Ying Zhuge, Longze Zhang, Theresa Cooley Zgela, Mary Sproull, Megan Mackey, Kevin Camphausen, Andra V. Krauze

**Affiliations:** 1Radiation Oncology Branch, Center for Cancer Research, National Cancer Institute NIH, 9000 Rockville Pike, Bethesda, MD 20892, USA; thomas.joyce@nih.gov (T.J.); sarisha.jagasia@nih.gov (S.J.); jasonshephard@gmail.com (J.S.); shreya.chappidi@nih.gov (S.C.); zhugey@mail.nih.gov (Y.Z.); longze.zhang@nih.gov (L.Z.); theresa.cooleyzgela@nih.gov (T.C.Z.); sproullm@mail.nih.gov (M.S.); mmackey@mail.nih.gov (M.M.); camphauk@nih.gov (K.C.); 2Department of Computer Science and Technology, University of Cambridge, 15 JJ Thomson Ave, Cambridge CB3 0FD, UK

**Keywords:** CD133, PROM1, serum, proteomics, biomarkers, glioblastoma

## Abstract

**Simple Summary:**

Glioblastoma (GBM) is an aggressive form of brain cancer characterized by its poor prognosis due to its resistance and recurrence capabilities. Liquid biopsies have been increasingly utilized when studying GBM as they provide extensive amounts of data at multiple time points without the invasiveness of tissue samples. We used a large-scale proteomic panel from the serum of GBM patients collected before chemoradiation therapy (CRT) to study the connections of CD133, a protein recognized as involved in GBM resistance. We used a novel machine learning process to identify 24 proteins associated with CD133 and 12-month survival, successfully grouping patients into risk groups based on protein profiles. Our results identify a potentially harmful protein profile while revealing important serum associations of CD133. These identified proteins are possible prognostic indicators and treatment entry points with heightened value because they can be frequently traced and monitored.

**Abstract:**

Glioma is the most prevalent type of primary central nervous system cancer, while glioblastoma (GBM) is its most aggressive variant, with a median survival of only 15 months when treated with maximal surgical resection followed by chemoradiation therapy (CRT). CD133 is a potentially significant GBM biomarker. However, current clinical biomarker studies rely on invasive tissue samples. These make prolonged data acquisition impossible, resulting in increased interest in the use of liquid biopsies. Our study, analyzed 7289 serum proteins from 109 patients with pathology-proven GBM obtained prior to CRT using the aptamer-based SOMAScan^®^ proteomic assay technology. We developed a novel methodology that identified 24 proteins linked to both serum CD133 and 12-month overall survival (OS) through a multi-step machine learning (ML) analysis. These identified proteins were subsequently subjected to survival and clustering evaluations, categorizing patients into five risk groups that accurately predicted 12-month OS based on their protein profiles. Most of these proteins are involved in brain function, neural development, and/or cancer biology signaling, highlighting their significance and potential predictive value. Identifying these proteins provides a valuable foundation for future serum investigations as validation of clinically applicable GBM biomarkers can unlock immense potential for diagnostics and treatment monitoring.

## 1. Introduction

Glioma is the most common type of primary central nervous system cancer, characterized by the uncontrolled growth of malignant cells within the brain or spinal cord. The most aggressive form of glioma is glioblastoma (GBM), which includes astrocytic tumors given a grade IV designation by the World Health Organization (WHO) [1]. GBM is defined by its poor prognosis, as its median survival is only 15 months, with a five-year survival rate of less than 5% [2]. There is not yet a cure for GBM. However, the standard of care (SOC) management involves maximal surgical resection followed by concurrent chemoradiation therapy (CRT) and adjuvant temozolomide (TMZ) [3].

The poor prognosis of GBM is due to the tumor’s resistance to treatment and the ability of certain cells within the tumor to evade therapies, particularly those with stem-like properties known as cancer stem cells (CSCs) [4]. CSCs are undifferentiated cells capable of sustaining tumor growth by generating both new CSCs and differentiated daughter cells [5]. Consequently, there is growing research interest in biomarkers that indicate the presence of CSCs due to their diagnostic and prognostic value.

CD133, or Prominin-1 (PROM-1), is a membrane-bound glycoprotein recognized as a CSC biomarker [5]. Recent research has shifted towards examining CD133’s role as a potential contributor to tumor resistance and recurrence in GBM [6]. Many of these studies have been conducted utilizing in vivo and in vitro methods [6]. However, these methods are limited as they are surrogate models rather than direct human research. Other studies have used tumor tissue at the time of initial resection or biopsy. However, the invasive nature of these methods makes them difficult to repeat over the course of a disease, preventing monitoring over time. 

This has spurred interest in using liquid biopsies, which involve blood, urine, or other bodily fluids as biospecimens. Liquid biopsies are cost-effective, minimally invasive, and can be collected frequently, reducing patient impact, and facilitating large-scale data acquisition. The extensive datasets generated, particularly from proteomic and metabolomic panels, have driven the adoption of machine learning (ML) to extract valuable insights [7,8,9]. An overarching goal of analyzing these large-scale panels is to identify and validate biomarkers that can indicate cancer presence or monitor treatment progress [10,11].

Survival is an important outcome endpoint in GBM as it is accurately calculated from the date of diagnosis to the date of death in most studies and registries, as compared to progression-free survival, which is subject to limitations in interpretation [12]. Assessment of survival at 12 months following diagnosis indicates the efficacy of treatment following SOC, given 6 to 12 months of adjuvant TMZ post-CRT. This time point indicates the disease trajectory thus rendering it a common clinical trial endpoint that can be compared across studies and with real-world data [2,13,14]. 

Although serum data have been explored in GBM [10,15,16,17], there remains a significant shortage of comprehensive large-scale data available. In this study, we utilized a large-scale proteomic panel based on serum collected from patients with pathology-proven GBM to investigate CD133. Our objective was to identify proteins related to CD133 using a clinically relevant survival endpoint. As a result, using ML, we identified proteins that were significantly associated with both CD133 and 12-month overall survival (OS). This approach combined correlation and feature selection (FS) analyses with survival and clustering evaluations for greater understanding. With further validation, our findings can enhance the understanding of the interplay between CD133 and CSCs, leading to improved prognosis, treatment monitoring, and optimization for better outcomes. The technological and clinical innovations of this study are summarized below.

### 1.1. Technological Innovations

This study is the first to develop an approach to large-scale serum proteome data based on a single literature-validated protein (CD133) and a clinically meaningful endpoint (12-month OS).The combined use of Pearson’s correlation analyses and Recursive Feature Selection, Cross-Validated (RFECV) for selecting serum proteins related to serum CD133 and 12-month OS in this study is novel.To ensure inclusive protein selection, we tested multiple FS methods on two base models with differing approaches to 12-month OS prediction.To validate our findings, we utilized two inverse but complementary processes involving correlation and FS to address the lack of large-scale data available for external validation.We used predicted hazard scores from the Cox Proportional Hazards Model for Gaussian Mixture Model clustering to categorize patients into risk groups based on their protein profiles.

### 1.2. Clinical Innovations

This is the first study to investigate CD133 and its associations using serum proteome data.The findings enhance the understanding of conclusions derived from serum data while addressing whether these conclusions are transferable with other types of biospecimens.Almost all identified proteins are either expressed in the brain, involved in brain function, and/or related to cancer biology pathways, underscoring their likely involvement in, or alteration from, GBM.The results identify potentially harmful protein profiles that might predispose individuals to short survival with SOC management, warranting further validation.The list of identified proteins offers candidates that can potentially be used to monitor prognosis and/or treatment as new data emerge and reference ranges are validated.

## 2. Materials and Methods

### 2.1. Dataset

Serum biospecimens collected from 109 patients with pathology-proven GBM (2005–2023) treated with CRT on National Institutes of Health (NIH) Institutional Review Board (IRB) approved studies before and after completion of CRT were included in the analysis. Serum samples were obtained prior to initiation of upfront CRT (mean 6.7 days, range (0 to 24)), then frozen at −80 °C for an average of 3951 days (range 239–7072 days), and finally thawed and screened using the aptamer-based SOMAScan^®^ proteomic assay technology. This multiplexed, aptamer-based approach (SomaScan^®^ assay) in the SomaLogic^®^ research facility used samples provided by the authors to measure the relative concentrations of 7596 protein targets (7289 human) for changes in expression using approximately 150 µL of serum [18,19]. The NIH Integrated Data Analysis Platform (NIDAP) [20] was employed for storage, query operations, and analysis. 

### 2.2. The Proposed Scheme

For the preliminary portion of this analysis, a combined evaluation utilizing Pearson’s correlation and machine learning (ML) in the form of FS was developed to identify proteins related to both CD133 and 12-month OS. This was performed using two parallel analyses: Process 1 and Process 2. Process 1 conducted the correlation analysis first and followed this with an FS analysis of the remaining proteins to select those most valuable to the prediction of 12-month OS. Process 2 first conducted the FS survival evaluation and then performed the correlation analysis on the resulting proteins. The final protein list included only those proteins that emerged from both processes, ensuring robust associations of the identified molecules with serum CD133 and 12-month OS. 

This final list of identified proteins was then utilized in the second portion of this analysis which began with using the Cox Proportional Hazards Model to further understand the individual protein contributions to 12-month OS. This analysis also produced predicted hazard scores for each patient that were used in Gaussian Mixture Model clustering to group patients into risk groups based on their protein profiles. The full process is summarized in Figure 1, and all code used in these analyses can be found in the Supplemental Materials.

### 2.3. Pearson’s Correlation Analysis

Pearson’s correlation analyses were used to identify pairwise linear relationships between the expression levels of panel proteins and the expression levels of CD133. This method was chosen due to its universality in research, its direct explainability, and the suitability of the shape of the data following log transformation [21]. Upon executing the analysis in both processes, we eliminated any proteins with a coefficient with an absolute value of less than 0.35 from the datasets, as this is an accepted lower boundary for moderate associations [22]. False discovery rate (FDR) was also integrated into this analysis portion as proteins with an FDR greater than 0.1 were simultaneously removed. We chose an FDR of 0.1 to allow for a wider inclusion of potential signals in the FS step of Process 1, as it increased the number of proteins from 161 to 216. This threshold was then carried over into Process 2.

### 2.4. Base Models

Two differing base models were employed for the FS portion of these processes to enhance protein selection. The first was Logistic Regression (LR), a supervised ML model typically used in binary classification. This model assigns coefficients to each variable and uses them to make classifications with a certain confidence level [23]. The other base model was Random Forest (RF), which utilizes decision trees rather than coefficients to make its classifications. Through recursive partitioning anchored in randomization, RF creates thresholds for different variables that lead down various paths, with outcomes being the pooled consensus of these pathways [24]. 

Each model was trained on the training datasets of both the correlated proteins of Process 1 and the full protein dataset used in Process 2. The log-transformed version of these data were utilized for all analyses, and scaling was performed using *MinMaxScaler()* from *sklearn.preprocessing* [25]. Additionally, a train–test split of 80% to 20% was utilized throughout. Optimal parameters for the base models were found using the *GridSearchCV()* package from *sklearn.model_selection* [26]. The penalty was kept constant at *l2* for LR, while the solvers were chosen from [*lbfgs*, *liblinear*, *sag*, *newton-cg*, *saga*], and the C values were picked from [*0.01*, *0.1*, *0*, *1*, *10*, *100*]. For RF, the max_depth was decided between (*3*, *6*, *9*, *None*), the max_features were selected from [*log*, *sqrt*], the max_leaf_nodes were chosen from (*3*, *6*, *9*, *None*), and the n_estimators were determined from (25, 50, 75, 100, 125, 150). 

For Process 1, the optimal parameters for LR were [solver= *liblinear*, C = 0.1], while for RF, they were [max_depth = 3, max_features = *log2*, max_leaf_nodes = 6, n_estimators = 125]. For Process 2, the optimal parameters for LR were [solver = *lbfgs*, C = 1], and for RF were [max_depth = 6, max_features = *log2*, max_leaf_nodes = *None*, n_estimators = 100]. 

Due to the imbalance of survivors and non-survivors in the training dataset, measures were taken to account and adjust for this. Following this fine-tuning, tests were carried out to determine whether the training data’s stratification helped improve the base models’ performance in either process. Improvements were only seen in the LR models of Process 2, so we only incorporated stratification in these tests. Additionally, LR models can incorporate “class_weights” as a parameter, which can be used to counteract specific imbalances through weight assignment, so specific weights were calculated and incorporated. For the LR models in Process 1, the weights were [0:0.613, 1:2.719], with 0 representing survivors and 1 representing non-survivors. Due to the stratification of the training data in Process 2, the weights changed slightly for this iteration to [0:0.630, 1:2.417].

### 2.5. Feature Selection Methods

Two FS methods with differing approaches were tested; however, only one was chosen for the full analysis. The first was the Least Absolute Shrinkage and Selection Operator (LASSO), which is known for its success when working with highly dimensional datasets. Using regularization, LASSO assigns and shrinks coefficients if unrelated to the variable of interest. By restraining the sum of these coefficients, many are shrunk to 0, which is how FS occurs [27].

The other FS method was RFECV, a wrapper-style FS model that initially classifies using all features and eliminates features in steps. After observing the initial performance of the model using all features, it ranks these features in terms of their importance to the model’s ability to classify. Then, at each step, it eliminates a subset and reevaluates the model’s performance while updating the rankings. This approach also accounts for the base model type, allowing for specialization in the optimal production of various models [28]. 

The effectiveness of the FS methods was evaluated using accuracy, Receiving Operator Characteristic (ROC)-Area Under the Curve (AUC), and Precision-Recall AUC. ROC-AUC and Precision-Recall AUC were utilized alongside accuracy to contextualize the model’s performance as they provide greater insight into the actual functioning and decision-making of the model. ROC-AUC focuses more on the classification of the survivors in this analysis, while the Precision-Recall AUC focuses on the smaller, non-survivor class as it is known for its value when working with imbalanced binary classification [29,30]. As a result, the three metrics together provided a holistic view of the model’s performance. 

Utilizing the base models and these performance metrics, LASSO and RFECV were evaluated using both datasets, and their accuracy scores are displayed in Table 1. The *RepeatedStratifiedKFold()* package from *sklearn.model_selection* was used for cross-validation across all tests to prevent overfitting [31]. Its parameters involved a split of 10 with 3 repeats at each step. Despite similar performance on the first dataset, the LASSO FS resulted in a reduction in accuracy from 0.82 to 0.73 when tested on the second dataset using LR, which was accompanied by decreases in ROC-AUC and Precision-Recall AUC. As a result, RFECV was chosen for this analysis.

Following this selection, RFECV was used in Processes 1 and 2 in their totality. RFECV allows for the modification of its “step” function, which involves the number of proteins eliminated at each pass, so to further improve its performance, various thresholds were tested. For Process 1, step values ranging from 1-5 were tried and selected due to the smaller dataset, and it was a step of 2 for LR and 3 for RF that performed best. For Process 2, due to the much larger set of values, step values between 10 and 50 were tested at increments of 10. For LR, a step of 20 was chosen to obtain the best results, while for RF, it was a step of 10. Finally, RFECV accounts for the base estimator, so LR and RF were tested separately, and the protein sets that they produced were combined. This was performed in both processes, and a final list of proteins was compiled aggregating overlapping proteins.

### 2.6. The Cox Proportional Hazards Model

After completing the two processes, a Cox Proportional Hazards Model analysis was conducted to contextualize the final protein list by elucidating the individual contributions of each protein to 12-month OS prediction. The *CoxPHFitter()* package from *lifelines* was used to produce parameter estimates, hazard ratios, and *p*-values [32]. Parameter estimates were evaluated based on their direction, magnitude, and accompanying hazard ratios. Selection of the most significant proteins was performed using parameter estimates and *p*-values and visualized as a volcano plot using IPA [33].

Given that these proteins were identified as major contributors to the prediction of 12-month OS, clustering of the predicted hazard scores for individual patients produced by the *CoxPHFitter()* package was utilized to group protein profiles associated with risk of death. Equation (1):(1)$$\beta^{T}X=\beta 1X1+\beta 2X2+\cdots +\beta pXp,$$
shows how these scores were calculated, where β represents the parameter estimate that was assigned to the protein, and *X* represents the actual expression value of that protein for an individual. For each patient, the parameter estimate for the protein was multiplied by the respective protein expression value, and this was performed for all proteins and then summed into one vector [34]. As shown by Equation (2):(2)$$h\left(t\right|X)=h_{0}(t)exp(\beta^{T}X),$$
the exponential function of this vector was then taken, meaning that the further this result deviated from 1, the greater the risk of that patient’s protein expression profile. At the same time, if this value was less than 1, this meant that the patient’s profile was protective. Finally, this value was multiplied by the baseline hazard ($h_{0}(t)$), which is the same for all patients, to account for the specified time point (12 months) [34].

### 2.7. Gaussian Mixture Model Clustering

After producing predicted hazard scores for each patient, Gaussian Mixture Model clustering was performed to group patients based on their protein profiles. Gaussian Mixture Model clustering was chosen because it is more flexible than traditional k-means clustering. Rather than simply calculating the distance from a centroid, it searches for various Gaussian distributions, which involves considering the variance of the data. Selection of the optimal number of clusters was performed utilizing Bayesian information criteria (BIC) scores, with the minimum score selected [35].

Using this clustering method, individuals with similar predicted hazard scores were grouped together. Given that the parameter estimates used in the calculation of the predicted hazard score for each patient were the same, this meant that patients in the same group had similar protein expression level values across proteins, and therefore, similar protein profiles. As a result, these clusters represented varying risk groups as predicted hazard scores are directly related to 12-month OS. This allowed for a survival analysis using Kaplan–Meier curves. It also resulted in visualizations of the protein profiles of each risk group to provide insight into which expression levels and patterns are possibly harmful.

## 3. Results

### 3.1. Correlation and Feature Selection Analysis

The breakdown of the protein elimination at each step of the processes is displayed in Figure 2. In Process 1, the 7289 original proteins were narrowed down to 216 proteins following the correlation analysis, of which 36 were chosen as most essential to 12-month OS prediction by RFECV. In Process 2, RFECV utilized 5120 proteins to predict 12-month OS, and 151 of these met Pearson’s correlation criteria. There was an overlap of 24 proteins, which comprised the final protein list, shown in Table 2. Access to the protein lists produced by each process (36 and 151) along with the final protein list (24) can be found in the Appendix A.

All models showed improvement after FS with increases in accuracy, ROC-AUC, and Precision-Recall rates in both base models in Process 1 and in the RF model of Process 2 (Figure 3). While the accuracy of the LR model in Process 2 did not increase, the model still showed improvement through higher ROC-AUC and Precision-Recall AUC values. Although the LR models outperformed the RF models in absolute metrics, the RF models demonstrated the most significant improvements in accuracy (59% to 73%) and Precision-Recall AUC (0.15 to 0.41). An LR model achieved the strongest improvement in ROC-AUC (from 0.67 to 0.79). Given the imbalanced training data, with 18 non-survivors compared to 69 survivors, the enhancements displayed in Precision-Recall AUC across these models indicated improved performance in detecting non-survivors, particularly with the RF models.

### 3.2. CD133 Serum

Pre-CRT serum CD133 expression levels measured in Relative Fluorescence Units (RFUs) for all 109 patients are displayed in Appendix A. Levels ranged from 13100 RFUs to 3326 RFUs (mean of 4982 RFUs) with CD133 having the 6th highest levels compared with the 24 identified proteins in the analysis. The highest protein expression was observed for PTPRS (mean of 61085 RFUs), which for clinical context was significantly closer to the expression level of serum Albumin (Appendix A). The time the samples were stored in the freezer from collection to analysis did not reveal any systematic impact on protein measurements, as shown in (Appendix A). 

### 3.3. The Cox Proportional Hazards Model

The results of the Cox Proportional Hazards Model indicate that proteins with parameter estimates of the highest magnitude are most impactful as seen in RPA2 (−6.67), ITGA6 (−5.27), and PDCL2 (−4.14). To identify the most significant contributors to 12-month OS, a volcano plot using the parameter estimates and *p*-values was created, shown in Figure 4a. The most significant proteins (lowest *p*-values) were RPA2 (parameter estimate = −6.67, HR = [0–0.76], *p* = 0.04) and AMPD2 (parameter estimate = −2.42, HR = [0.01–1.00], *p* = −0.05). TRPA2, AMPD2, DLK2, NEGR1, POLI, ITGA6, CLN5, P3H1, PDCL2, and CEACAM3 were the 10 proteins with the most significant contributions (Figure 4A) in the complete Cox Proportional Hazards Model analysis (Appendix A).

Heat maps for each of the identified molecules were created and visualized, contextualizing patterns of serum expression levels in relationship to serum CD133. Patterns involving expression levels of these proteins with CD133 were evident and expected given the correlation component of this analysis. Of the 10 proteins identified, 6 generally aligned with the low to high expression level pattern of CD133 compared to 3 that had inverse patterns, while AMPD2 had a more ambiguous signature (Figure 4b).

### 3.4. Gaussian Mixture Model Clustering

The Gaussian Mixture Model clustering that produced the lowest BIC score using the predicted hazard scores resulted in five survival risk groups, labeled Lowest Risk Group (*n* = 20), Low Risk Group (*n* = 18), Medium Risk Group (*n* = 49), High Risk Group (*n* = 16), and Highest Risk Group (*n* = 6). The Kaplan–Meier survival analysis for 12-month OS was statistically significant (*p* < 0.0001) (Figure 5a) and revealed differences in survival (Figure 5b) with the Lowest Risk Group having 100% survival at 12 months, while the Highest Risk Group had 0% 12-month survival.

Alongside this survival exploration, the complementary aim of the clustering analysis was to evaluate the protein profiles of individuals in each survival group to identify patterns of protein expression (Figure 6). Several proteins had opposite expression levels in the Highest Risk vs. Lowest Risk groups. CEACAM3, SAT2, TRAPPC5, PDCL2, RPA2, ITGA6, CSNK2B, and Furin all have their highest mean expression levels in the Lowest Risk group with their lowest mean expression levels in the Highest Risk group. SELENOW, TIMM8A, and PCDHGA10 revealed the opposite, with their lowest mean values in the Lowest Risk group and their highest mean values in the Highest Risk group. More variation was identified in the middle three groups, and some seemingly contradictory results between the Lowest Risk Group and the Low Risk Group, as well as between the Highest Risk group and the High Risk group were also seen. However, the table indicates that extremely high or low mean protein expression levels seem to have the most significant impact on 12-month OS.

## 4. Discussion

CD133 has primarily been used as a biomarker for identifying CSCs, initially gaining recognition for its potential diagnostic and prognostic potential [36,37,38]. However, in recent years, its role has shifted towards being considered a functional unit in tumor resistance and recurrence, as explored in our previous review [6]. Serum data have gained attention in GBM biomarker research due to its ability to evaluate samples throughout the course of a disease noninvasively. Consequently, using a large-scale proteomic panel derived from patients with pathology-proven GBM, this analysis demonstrated that CD133 can be effectively measured in serum, enabling the investigation of its potential utility and associations with other markers and survival outcomes. Although several prior studies have examined serum proteins in GBM [39,40,41,42], this is the first to focus on CD133 specifically. We achieved this by developing a novel multi-step ML process that identified proteins associated with both CD133 and 12-month OS, whose validity was confirmed through survival analyses and clustering. Although previous studies have utilized ML to identify GBM biomarkers and analyze omic panels, the selected proteins often lack direct clinical relevance, connections to known prognostic markers, and/or a clear biological mechanism. By centering this investigation on a known biomarker and a well-defined outcome endpoint (12 months), we identified proteins relevant to resistance and recurrence with a shared connection, enhancing potential clinical applicability.

Elevated CD133 in tumor tissue has largely been associated with poor prognosis and survival [38,43,44,45], though some studies have reported non-contributory CD133 cells that do not support this trend [46]. In this study, serum CD133 expression varied across the population without a clear correlation to 12-month patient survival. This suggests that signals from both dead and living cells might be captured. This finding is crucial for understanding the transferability of serum data-derived conclusions compared to other biospecimens, emphasizing the need for ongoing research and validation. Additionally, several of the identified final proteins showed higher expression levels than CD133, with PTPRS being the highest, indicating the potential for other proteins to serve as proxies for CD133 or its associations in future studies.

Due to the present lack of transferability across biospecimens, there are limited data available for external validation [47,48] with no serum proteomic data available in GBM for direct comparison. To address this, we aimed for enhanced internal validation utilizing two coinciding processes, resulting in the final protein list. Previous studies have utilized statistical analyses alongside FS [49,50], with multiple using Pearson’s correlation coefficients in particular prior to or following FS [51,52]. Our novel approach used both which, combined with the optimization of the base models and FS methods, ensured that the selected proteins had robust associations with CD133 and 12-month OS. The effectiveness of this methodology was confirmed through analysis using the Cox Proportional Hazards Model paired with Gaussian Clustering. Similar to Liu et al. (2023), the Cox analysis was employed following FS to provide further insight into the individual survival contributions of the proteins in relation to the group. Combined FS and Cox analyses, including LASSO-Cox and Elastic-Net with Cox, have become popular ways to narrow and rank expansive datasets [53,54,55,56,57]. The Cox analysis identified two proteins (RPA2 and AMPD2) with statistically significant results, while the other important contributors were determined based on parameter estimates and *p*-values.

A clustering analysis was completed that separated patients into five survival risk categories based on their distinct protein profiles [58,59]. The statistically significant survival separation between the survival groups reinforced the clinical value of the final proteins while justifying the utilization of CD133 as the basis for this study. The relative mean expression value comparisons, suggested that the Highest Risk category may be harboring a harmful serum protein profile, irrespective of treatment, as none of these individuals survived to 12 months. This implied a potential predisposition to short survival, which, through future validation, may reveal that CRT is ineffective in overcoming. As a result, targeted interventions based on manipulating these protein levels or upstream mediators may allow for prolonged survival. Conversely, the Lowest Risk group achieved 100% 12-month OS, indicating a possible protective nature to this expression pattern.

Several of the proteins identified in this analysis were found to be expressed in the brain, associated with a specific role in brain function, and/or pertinent to tumor biology (Figure 7a) [60]. This notably included RGS4 (nearly exclusively expressed in the brain) [61], PCDHGA10 (most abundant in the brain) [62,63], NEGR1 (involved in synaptogenesis and neuronal arborization) [64,65], DLK2 (EGF-like Notch inhibitor associated with neuronal differentiation and stemness) [66,67] and SCN3B (associated with ion channel transport, immune response, and glioma-related epilepsy) [68]. The expression of these molecules has shown nearly exclusive or predominant neural expression [60], which provided a form of external validation of the proteins identified in this analysis, while further reinforcing the value of CD133 as the central protein and revealing the potential utility of these proteins when captured in serum (Figure 7b). Expression levels of DLK2 and IL15RA were also aligned across survival risk groups with IL15RA being relevant given its evolving role in targeted therapies to overcome resistance and improve anti-tumor effect [69]. The observed relationships implied a potential connection between CD133, CSCs, and immune and inflammatory responses that can be captured in GBM serum. CREB3L1 is expressed in the brain but is also a prominent transcription factor in cancer biology [70] that has been studied as a biomarker in glioma, given its association with tumor grade and survival [71].

Although the identified proteins originate from various subcellular locations and tissues of origin [60], they share associations with several biological pathways relevant to GBM, including replication and repair, metabolism and energy currency, neural development, and immune response (Figure 8). RPA2 and POLI are associated with DNA replication and repair and have recently been linked to TMZ and chemoresistance in GBM [74]. RPA2, POLI, and CSNK2B share the nucleus as a subcellular location, which may be relevant for the determination of whether dead cells are being measured. CSNK2B is also connected to cell signaling pathways, alongside ITGA6 and Furin, with documented contributions to neurodevelopmental disorders, epilepsy, and the classical GBM subtypes [75,76]. ITGA6 has also been associated with GBM subtypes and DNA damage response [77], while Furin is a component of secretory pathways connected with tumor immunity in various cancers [78]. In the survival clustering, CSNK2B, ITGA6 and Furin had their highest mean expression level in the Lowest Risk group and their lowest in the Highest Risk group.

Important connections to metabolism and energy currency also involved AMPD2 and SAT2, which are associated with purine and polyamine metabolism [79,80]. Specifically, SAT2 is altered in various malignancies and associated with tumorigenesis and radiation response [81,82,83]. Further, SELENOW is expressed in the brain and is implicated in EGF signaling and redox regulation [84], while TIMM8A is involved in mitochondrial function with its role in cancer currently evolving [85]. PTPRS is significant in that it can distinguish CSCs from normal brain cells using single-cell RNA sequencing [86] and is one of seven genes whose mRNA expression is associated with tumor grade, with GBM exhibiting the highest levels [87]. In this analysis, however, serum PTPRS expression did not effectively associate with survival risk groups. Additional signals included TRAPP5 and CEACAM3, the roles of which are currently being determined in cancer and GBM. Comparisons were also made with the most relevant literature proteins compiled in our previous review [6]; however, no specific shared proteins were identified. This further reinforced that serum connections may not translate to other biospecimen relationships. This is of particular relevance since currently there is no serum based GBM dataset available for comparison and comparison with proteomic datasets that are not originating from serum may produce discordant results in some signals depending on their subcellular compartment of origin or disease entity [88]. 

Limitations of this study include the long timeframe over which data collection occurred and the lack of independent serum data for comparison. We have, however, examined the captured serum signal of several proteins, including CD133, in relationship to the time from sample collection to analysis and have not observed any systematic impact on protein measurement [89,90]. Serum data may also have different ranges as this is an emerging field with various collection methods and measurement technologies. Some of the groups produced in the clustering analysis resulted in varying numbers of patients, some of which were low (e.g., highest risk group n = 6) and this aspect limits applicability without large-scale data validation. Thus, conclusions should be drawn with caution. Given the intricacy of interpretation for protein signals in serum as belonging to dead or living cells there is significant intersectionality with clinical outcomes. As large-scale serum proteomic panels grow, several aspects will improve interpretation: (1) co-measurement of large-scale proteomic signals in tissue and in serum in the same patient at the same or similar timepoint; (2) correlation with other data streams such as circulating tumor cells with viability staining; (3) comparison of large-scale proteomic signatures with normal volunteer samples. While these data are currently lacking, apoptosis or necrosis markers may provide proxy signals for the detection of cell death. This last point is subject to future directions using the current dataset as is investigating the alteration of serum proteins with treatment. In addition, the performance and interpretability of the current approach need to be explored in additional datasets, local and public, and this work is ongoing. The emergence of comparable serum panels would allow for external validation and the ability to identify clinically applicable proteins more strongly. It would also allow for the publication of values and ranges that could be investigated in clinical and laboratory settings. However, this is contingent on the continued exploration of serum data, and although there are over 30 GBM trials currently occurring, the majority are studying tissue [88]. Serum analyses have the potential to provide clinicians with more patient information through non-invasive methods, but this depends on the continued development of this field over the next decade.

## 5. Conclusions

This research introduced an innovative method for investigating the relationships and survival implications of a specific protein using a serum proteomic panel. By focusing on CD133, this study revealed how this protein behaves in serum and identified other proteins associated with both CD133 and 12-month OS. These findings suggest that CD133 may play a role in tumor resistance and recurrence and that its influence can be captured in serum. Additionally, this study categorized survival risks based on protein profiles, shedding light on potentially harmful protein expression combinations that could lead to shorter survival despite CRT.

As proteomic serum data collection continues to expand and external validation becomes more feasible, tracking GBM prognosis and treatment effectiveness using non-invasive serum samples becomes increasingly likely. Comparing independently collected panels would allow us to refine ranges for these proteins, ultimately determining their clinical applicability. This study’s identified proteins serve as a crucial starting point for future research that, once further built upon, have expansive clinical potential. 

## Figures and Tables

**Figure 1 cancers-16-02740-f001:**
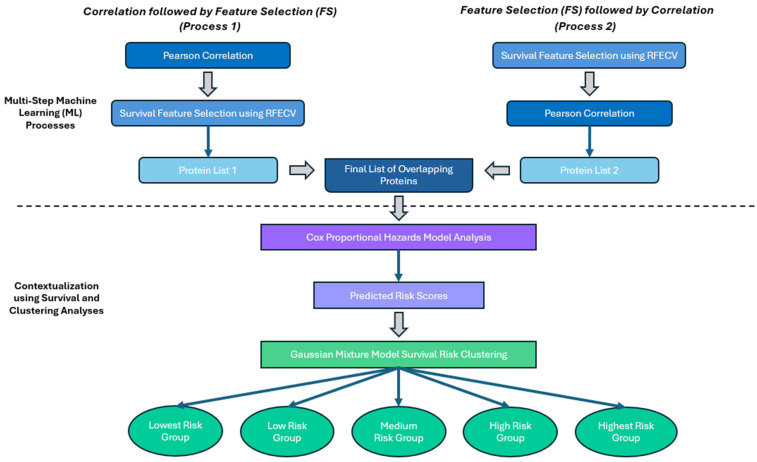
General overview of the proposed methodology for protein selection and analysis.

**Figure 2 cancers-16-02740-f002:**
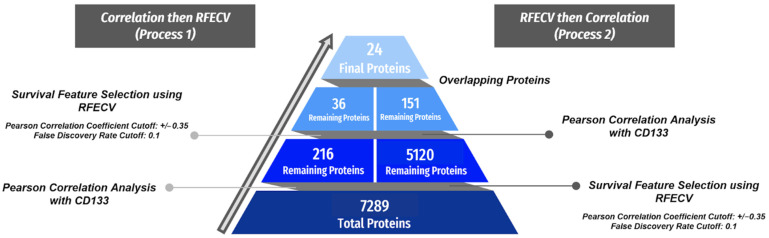
Summary of the initial machine learning data process with protein values. Two parallel processes were employed with overlapping proteins included in the final protein set. The left side shows the progression of Process 1, in which Pearson’s correlation analysis was completed first, followed by FS using RFECV at 12-month OS. The right shows Process 2, which followed an inverse but identical trajectory starting with FS and following this with the correlation analysis. For Pearson’s correlation analysis, proteins with a correlation coefficient with an absolute value greater than 0.35 and a false discovery rate less than 0.1 were maintained.

**Figure 3 cancers-16-02740-f003:**
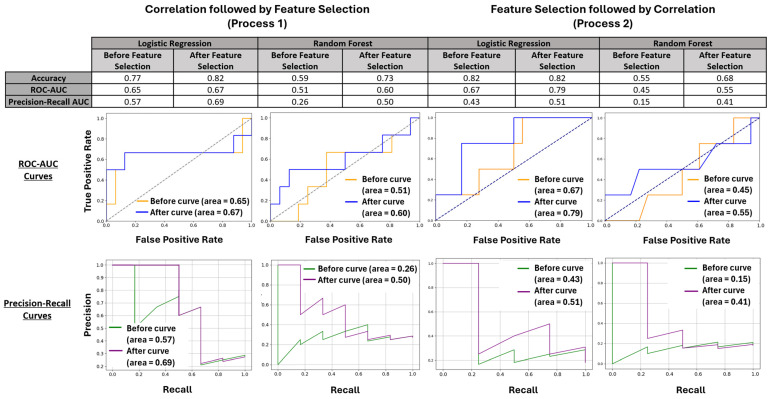
LR and RF were utilized in both iterations of this process and were evaluated using accuracy, ROC-AUC, and Precision-Recall AUC. The initial tables summarize the improvements made using these processes accompanied by ROC-AUC curves and Precision-Recall AUC curves that both visualize and contextualize these values.

**Figure 4 cancers-16-02740-f004:**
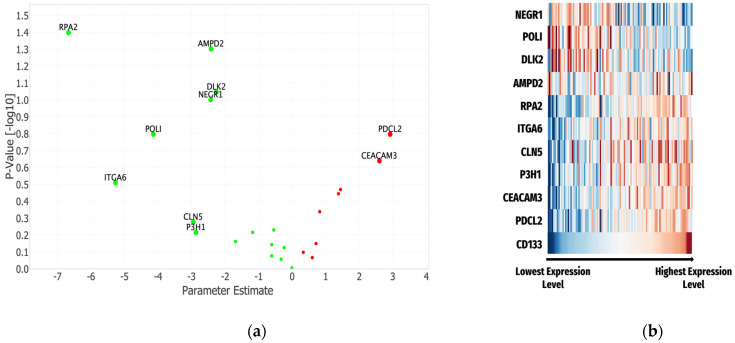
(**a**) Volcano plot of the Cox Proportional Hazards Model analysis for the proteins identified as connected to serum CD133 and 12-month OS. Top 10 proteins based on *p*-values and parameter estimates were labeled. (**b**) Heat map of log transformed serum protein expression levels displayed with CD133 expression for comparison. The median values for each individual protein were employed with each band representing the value for each individual patient. (Red to orange—indicates protein expression values greater than the median, Blue to light blue—indicates expression levels less than the median, both becoming darker with movement away from the median).

**Figure 5 cancers-16-02740-f005:**
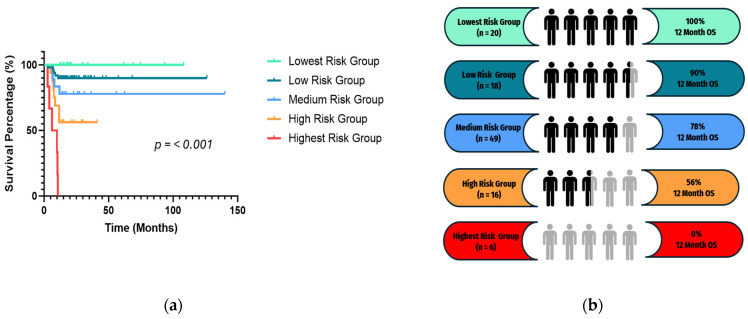
(**a**) Survival analysis of the clustering groups based on 12-month OS. (**b**) Summary of the 12-month OS rates in each risk group represented on the survival graph with corresponding colors.

**Figure 6 cancers-16-02740-f006:**
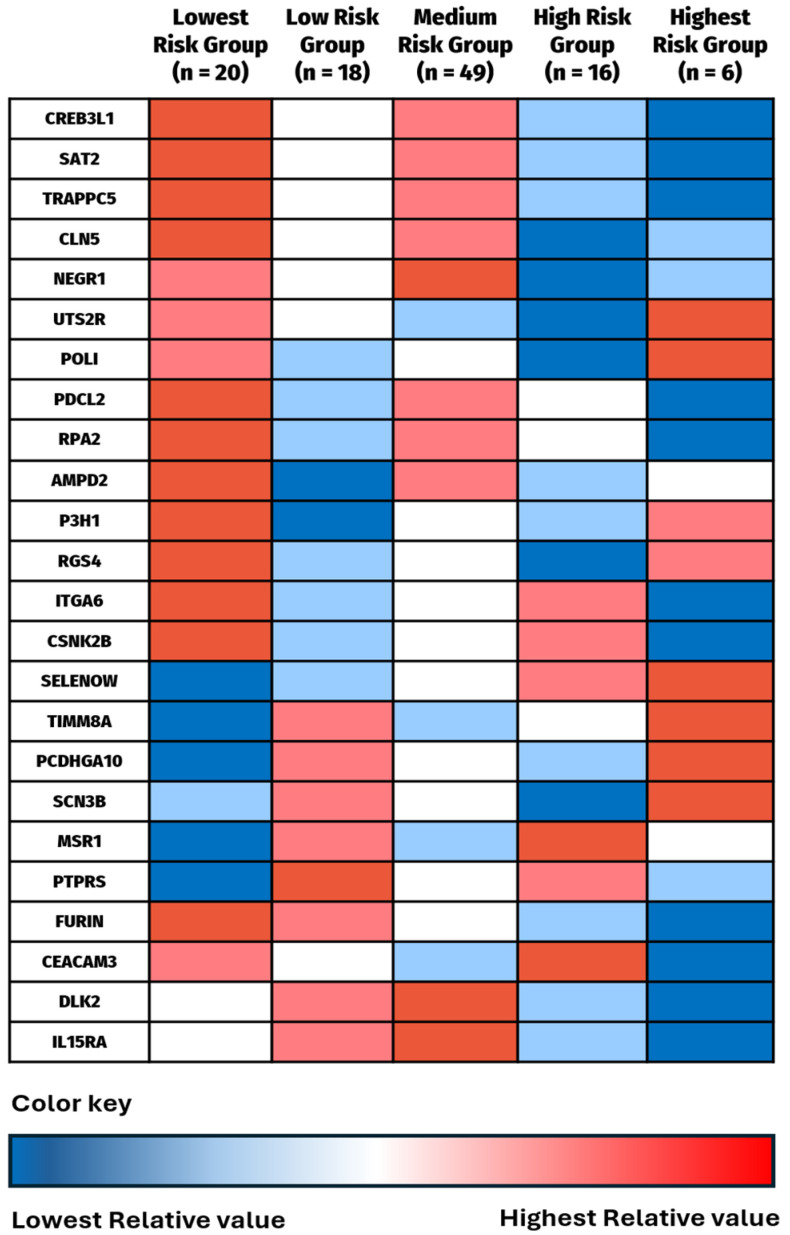
Comparison of mean expression values for each identified protein by survival risk group. The group with highest mean value was given the darkest red color, the second highest value lighter red, the middle value white, the second lowest value light blue, and the lowest value dark blue. This color gradient was chosen to stay consistent with the previous heat maps in Figure 4b. Numerical data are shown in Appendix A.

**Figure 7 cancers-16-02740-f007:**
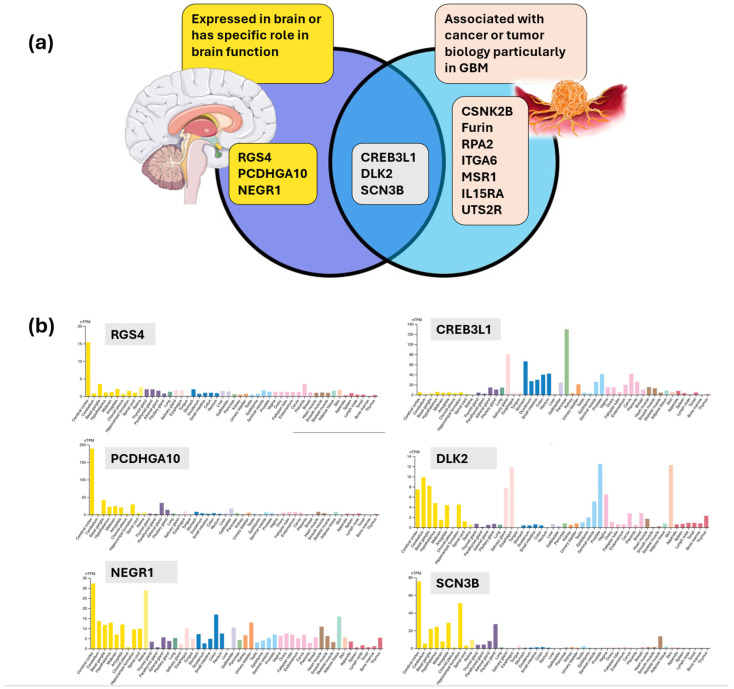
(**a**) Serum CD133 is associated with molecules expressed in the brain and associated with tumor biology, particularly in GBM [72,73]. (**b**) Human protein atlas RNA expression overview profiles for RGS4, PCDHGA10, NEGR1, CREB3L1, DLK2, and SCN3B illustrating tissue specificity with a predominance of association in brain tissue (yellow bars) [60].

**Figure 8 cancers-16-02740-f008:**
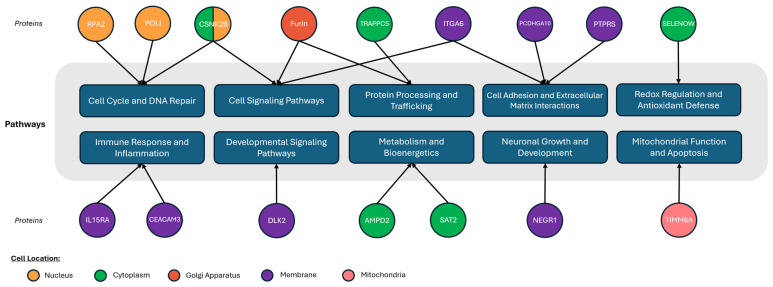
Subcellular location and pathway linkages for the identified proteins [74,75,76,77,78,79,80,81,82,83,84,85,86,87].

**Table 1 cancers-16-02740-t001:** Accuracy rates from the RFECV and LASSO FS methods employed to select the optimal approach for protein selection. Decreased accuracy when using LASSO in Process 2 resulted in the selection of RFECV for analysis.

		Correlation Followed by Feature Selection (FS) (Process 1)		Feature selection (FS) Followed by Correlation (Process 2)	
		Before FS	After FS	Before FS	After FS
**Logistic Regression**	**LASSO**	0.77	0.77	0.82	0.73
	**RFECV**	0.77	0.82	0.82	0.82
**Random Forest**	**LASSO**	0.59	0.73	0.55	0.68
	**RFECV**	0.59	0.73	0.55	0.68

**Table 2 cancers-16-02740-t002:** List of proteins identified connected to serum CD133 and 12-month OS.

Entrez Gene Symbol	Target Full Name
RPA2	Replication protein A 32 kDa subunit
AMPD2	AMP deaminase 2
DLK2	Protein delta homolog 2
NEGR1	Neuronal growth regulator 1
PDCL2	Phosducin-like protein 2
POLI	DNA polymerase iota
CEACAM3	Carcinoembryonic antigen-related cell adhesion molecule 3
ITGA6	Integrin alpha-6
PCDHGA10	Protocadherin gamma-A10
SELENOW	Selenoprotein W
TIMM8A	Mitochondrial import inner membrane translocase subunit Tim8 A
CLN5	Ceroid-lipofuscinosis neuronal protein 5: Lumenal domain
IL15RA	Interleukin-15 receptor subunit alpha
P3H1	Prolyl 3-hydroxylase 1
UTS2R	Urotensin-2 receptor
RGS4	Regulator of G-protein signaling 4
PTPRS	Receptor-type tyrosine-protein phosphatase S
FURIN	Furin
MSR1	Macrophage scavenger receptor types I and II: Extracellular domain
CSNK2B	Casein kinase II subunit beta
SAT2	Diamine acetyltransferase 2
TRAPPC5	Trafficking protein particle complex subunit 5
CREB3L1	Cyclic AMP-responsive element-binding protein 3-like protein 1
SCN3B	Sodium channel subunit beta-3

## Data Availability

Responsible data sharing is part of our mandate and the cardinal future direction of this dataset. We are currently aggregating several data aspects including additional samples and omic data and once this is complete, the set will be de-identified and shared. The goal is to provide one comprehensive GBM dataset that has clinical, omic and imaging data. Individual requests to access the datasets should be directed to Andra V. Krauze.

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
