# Peer review of "Serum CD133-Associated Proteins Identified by Machine Learning Are Connected to Neural Development, Cancer Pathways, and 12-Month Survival in Glioblastoma"

_cancers, 2024, doi:10.3390/cancers16152740_

Round 1

Reviewer 1 Report

Comments and Suggestions for Authors

The study titled "Serum Proteome-Associated Proteins Identified by Machine Learning are Connected to Neural Development, Cancer Pathways, and 12-Month Survival in Glioblastoma" by Joyce et al. is a significant contribution to the field. The primary objective of the study was to examine the role of CD133 and its associations using serum proteome data in glioblastoma patients. The study focused on identifying proteins linked to CD133 and their correlation with the 12-month overall OS.

The findings hold considerable promise for enhancing our understanding of the interplay between CD133 and the poor prognosis of glioblastoma, potentially leading to improved treatment monitoring and optimization for better outcomes.

Notably, the study utilized a machine learning approach to select serum proteins associated with CD133 and 12-month OS, showcasing technological innovation. Furthermore, the identification of potentially detrimental protein profiles that may predispose individuals to shorter survival under standard care management represents a clinical innovation.

These findings could offer candidates for prognostic monitoring and treatment evaluation. The study also highlights the potential of non-invasive serum samples for prognostic tracking and treatment efficacy assessment, opening up exciting avenues for future research.

This study lays a robust foundation for further exploration and holds significant clinical potential. I strongly recommend this manuscript for publication in Cancers.

Author Response

Reviewer 1:

The study titled "Serum Proteome-Associated Proteins Identified by Machine Learning are Connected to Neural Development, Cancer Pathways, and 12-Month Survival in Glioblastoma" by Joyce et al. is a significant contribution to the field. The primary objective of the study was to examine the role of CD133 and its associations using serum proteome data in glioblastoma patients. The study focused on identifying proteins linked to CD133 and their correlation with the 12-month overall OS.

The findings hold considerable promise for enhancing our understanding of the interplay between CD133 and the poor prognosis of glioblastoma, potentially leading to improved treatment monitoring and optimization for better outcomes.

Notably, the study utilized a machine learning approach to select serum proteins associated with CD133 and 12-month OS, showcasing technological innovation. Furthermore, the identification of potentially detrimental protein profiles that may predispose individuals to shorter survival under standard care management represents a clinical innovation.

These findings could offer candidates for prognostic monitoring and treatment evaluation. The study also highlights the potential of non-invasive serum samples for prognostic tracking and treatment efficacy assessment, opening up exciting avenues for future research.

This study lays a robust foundation for further exploration and holds significant clinical potential. I strongly recommend this manuscript for publication in Cancers.

Response: We thank the reviewer for the time spent evaluating our paper and for their summary. Much appreciated.

Reviewer 2 Report

Comments and Suggestions for Authors

In this paper, the authors presents a method utilizing machine learning to analyze serum proteomic data from patients with Glioblastoma (GBM), a highly aggressive brain cancer. The study employed a SOMAScan®-based assay to profile 7289 serum proteins and developed a novel machine learning methodology to identify 24 proteins associated with both CD133—a known biomarker for GBM resistance—and 12-month overall survival. The findings categorized patients into five risk groups, providing insights into potential prognostic indicators and therapeutic targets, and highlighting the value of liquid biopsies for non-invasive monitoring of disease progression and treatment response. The following lists some comments for consideration.  Firstly, regarding data collection, particular attention should be given to the extended duration over which serum samples were procured, necessitating the introduction of detailed information pertaining to these patients and the conducted experiments. Secondly, the findings are grounded on a comparatively limited sample size for specific risk cohorts, potentially impeding the broad applicability of the results. To improve the credibility of identified biomarkers, it is advisable to employ independent serum datasets for external validation, a crucial step in affirming their reliability. Furthermore, it is essential that these data be accessible to the research community for academic endeavors. Thirdly, the interpretation of serum data is intricate, as it can encompass signals from both deceased and viable cells, thereby entangling the correlation with clinical outcomes. Additionally, the superiority and interpretability of the machine learning model employed need be rigorously confirmed.

Comments on the Quality of English Language

Minor English need be polished.

Author Response

In this paper, the authors presents a method utilizing machine learning to analyze serum proteomic data from patients with Glioblastoma (GBM), a highly aggressive brain cancer. The study employed a SOMAScan®-based assay to profile 7289 serum proteins and developed a novel machine learning methodology to identify 24 proteins associated with both CD133—a known biomarker for GBM resistance—and 12-month overall survival. The findings categorized patients into five risk groups, providing insights into potential prognostic indicators and therapeutic targets, and highlighting the value of liquid biopsies for non-invasive monitoring of disease progression and treatment response.

The following lists some comments for consideration. 

  1. Firstly, regarding data collection, particular attention should be given to the extended duration over which serum samples were procured, necessitating the introduction of detailed information pertaining to these patients and the conducted experiments.

Response: We agree with the reviewer. This is an important point. In the materials and methods section, we did acknowledge the prolonged period of time over which the samples were collected, and then in the discussion addressed it as a limitation. We agree that this needs to be enhanced and have now added the measured CD133 levels in each patient in relationship to the time the sample was stored in the freezer as Supplementary Figure 3 in the results section. We have further developed on this idea in the discussion to highlight that we have not noted any relationship between the time the samples were stored and measured levels of proteins and have added references to our previous work with these samples in previous analyses. Added text and supplementary figure below.

“The time the samples were stored in the freezer from collection to analysis did not reveal any systematic impact on protein measurements, as shown in (Supplemental Figure 3).”

Supplementary Figure 3. CD133 protein value based on the 7k proteomic panel in relation to the time the samples were stored in the freezer from collection to analysis in days showing the signal measured in the CD133 protein value pre (blue) vs. post CRT (magenta), and the number of days in the freezer for each patient (orange). The CD133 expression does not correlate with the number of days the samples were stored in the freezer.

“We have however examined the captured serum signal of several proteins including CD133 in relationship to the time from sample collection to analysis and have not ob-served any systematic impact on protein measurement [88, 89].”

  1. Krauze, A.V., et al., Glioblastoma survival is associated with distinct proteomic alteration signatures post chemoirradiation in a large-scale proteomic panel. Front Oncol, 2023. 13: p. 1127645.
  2. Krauze, A.V., et al., Revisiting Concurrent Radiation Therapy, Temozolomide, and the Histone Deacetylase Inhibitor Valproic Acid for Patients with Glioblastoma-Proteomic Alteration and Comparison Analysis with the Standard-of-Care Chemoirradiation. Biomolecules, 2023. 13(10).

  1. Secondly, the findings are grounded on a comparatively limited sample size for specific risk cohorts, potentially impeding the broad applicability of the results.

Response: We agree with the reviewer and have now emphasized this aspect in the discussion.

“Some of the groups produced in the clustering analysis resulted in varying smaller amounts numbers of patients, some of which were low (e.g. highest risk group n=6) and this aspect limits applicability without large scale data validation. Thus, conclusions should be drawn with caution.”

  1. To improve the credibility of identified biomarkers, it is advisable to employ independent serum datasets for external validation, a crucial step in affirming their reliability.

Response: We agree with the reviewer and share this concern as a critical aspect to advancing large scale data analysis in non-invasive specimens such as serum towards clinically actionable biomarkers. Currently there is no other serum data set in GBM that we can employ for comparison.

The ongoing work with this and similar sets thus needs to be disseminated to encourage more biospecimen acquisition, analysis and sharing of the emerging data. We have now highlighted this in the discussion and added a reference from one of our recent papers that illustrates this point.

“This is of particular relevance since currently there is no serum based GBM dataset available for comparison and comparison with proteomic datasets that are not originating from serum may produce discordant results in some signals depending on their sub-cellular compartment of origin or disease entity [88].”

  1. Tasci, E., et al., MGMT ProFWise: Unlocking a New Application for Combined Feature Selection and the Rank-Based Weighting Method to Link MGMT Methylation Status to Serum Protein Expression in Patients with Glioblastoma. Int J Mol Sci, 2024. 25(7).

  1. Furthermore, it is essential that these data be accessible to the research community for academic endeavors.

Response: We agree with the reviewer. Responsible data sharing is part of our mandate and the cardinal future direction of this dataset. We are currently aggregating several data aspects including additional samples and omic data and once this is complete, the set will be de-identified and shared. This is our plan. We wanted to provide one comprehensive GBM dataset that has clinical, omic and imaging data. Accomplishing this is challenging and time consuming but essential. We have now added this context to the Data Availability Statement.

  1. Thirdly, the interpretation of serum data is intricate, as it can encompass signals from both deceased and viable cells, thereby entangling the correlation with clinical outcomes.

Response: We agree with this critical point and have now commented on this aspect in the discussion. This is a challenging problem to solve. We have enhanced the discussion to add the following:

“Given the intricacy of interpretation for protein signals in serum as belonging to dead or living cells, there is significant intersectionality with clinical outcomes.  As large-scale serum proteomic panels grow, several aspects will improve interpretation: 1) co-measurement of large-scale proteomic signals in tissue and in serum in the same patient at the same or similar timepoint; 2) correlation with other data streams such as circulating tumor cells with viability staining; 3) comparison of large-scale proteomic signatures with normal volunteer samples. While this data is currently lacking, apoptosis or necrosis markers may provide proxy signals for the detection of cell death. This last point is subject to future directions using the current dataset.”

  1. Additionally, the superiority and interpretability of the machine learning model employed need be rigorously confirmed.

Response: We agree that this is the case. We intend on utilizing this and similar approaches in future datasets originating from this cohort as well as on public omic datasets to assess performance and validate/compare findings. Additional analyses are also ongoing aimed specifically at interpretability of the model. We have added this to the limitations of the study section in the discussion section.

“In addition, the performance and interpretability of the current approach needs to be explored in additional datasets local and public and this work is ongoing. “

Comments on the Quality of English Language

Minor English need be polished.

Response: We have made English language corrections and performed a Grammarly check resulting in a 95% overall score post the corrections made after addressing the comments.
